# Factors associated with pneumococcal carriage and density in children and adults in Fiji, using four cross-sectional surveys

Eleanor F. G. Neal[1,2]*, Cattram D. Nguyen[1,2], Felista T. Ratu[3], Eileen M. Dunne[1,2], Mike Kama[3], Belinda D. Ortika[1], Laura K. Boelsen[1], Joseph Kado[4,5], Lisi Tikoduadua[3], Rachel Devi[3], Evelyn Tuivaga[3], Rita C. Reyburn[1], Catherine Satzke[1,2], Eric Rafai[3], E. Kim Mulholland[1,2,6], Fiona M. Russell[1,2]

**1** Infection & Immunity, Murdoch Children's Research Institute, Royal Children's Hospital, Parkville, VIC, Australia, **2** Department of Paediatrics, The University of Melbourne, Parkville, Australia, **3** Ministry of Health and Medical Services, Suva, Fiji, **4** Telethon Kids Institute, University of Western Australia, Western Australia, Australia, **5** College of Medicine Nursing and Health Sciences, Fiji National University, Suva, Fiji, **6** Department of Infectious Disease Epidemiology, London School of Hygiene and Tropical Medicine, London, United Kingdom

* eleanor.neal@rch.org.au

**Data Availability Statement:** All relevant data are not publicly available because the ethics committee approved the study protocol, which specified the

## Abstract

This study describes predictors of pneumococcal nasopharyngeal carriage and density in Fiji. We used data from four annual (2012–2015) cross-sectional surveys, pre- and post-introduction of ten-valent pneumococcal conjugate vaccine (PCV10) in October 2012. Infants (5–8 weeks), toddlers (12–23 months), children (2–6 years), and their caregivers participated. Pneumococci were detected and quantified using *lyt*A qPCR, with molecular serotyping by microarray. Logistic and quantile regression were used to determine predictors of pneumococcal carriage and density, respectively. There were 8,109 participants. Pneumococcal carriage was negatively associated with years post-PCV10 introduction (global *P*<0.001), and positively associated with indigenous iTaukei ethnicity (aOR 2.74 [95% CI 2.17–3.45] *P*<0.001); young age (infant, toddler, and child compared with caregiver participant groups) (global *P*<0.001); urban residence (aOR 1.45 [95% CI 1.30–2.57] *P*<0.001); living with ≥2 children <5 years of age (aOR 1.42 [95% CI 1.27–1.59] *P*<0.001); low family income (aOR 1.44 [95% CI 1.28–1.62] P<0.001); and upper respiratory tract infection (URTI) symptoms (aOR 1.77 [95% CI 1.57–2.01] *P*<0.001). Predictors were similar for PCV10 and non-PCV10 carriage, except PCV10 carriage was negatively associated with PCV10 vaccination (0.58 [95% CI 0.41–0.82] *P* = 0.002) and positively associated with exposure to household cigarette smoke (aOR 1.21 [95% CI 1.02–1.43] *P* = 0.031), while there was no association between years post-PCV10 introduction and non-PCV10 carriage. Pneumococcal density was positively associated with URTI symptoms (adjusted median difference 0.28 [95% CI 0.16, 0.40] *P*<0.001) and toddler and child, compared with caregiver, participant groups (global *P* = 0.008). Predictors were similar for PCV10 and non-PCV10 density, except infant, toddler, and child participant groups were not associated with PCV10 density. PCV10 introduction was associated with reduced the odds of overall and PCV10 pneumococcal carriage in Fiji. However, after adjustment iTaukei ethnicity was positively

study objectives and how data would be used. These restrictions apply to all data included in this manuscript. It is not consistent with our ethical permissions to share de-identified or aggregate versions of our data, as publicly available data could be used for purposes that were not specified in the protocol approved by the ethics committee, and therefore would be a breach of our ethics permissions. During the informed consent, the purpose of the study was explained to participants, and they were told how their data would be used. The use of these data for a new purpose that was not included in the approved study protocol would require additional ethical approval from the Fijian National Health Research and Ethics Review Committee. Following approval, de-identified data would be made available. Additionally, this process is mindful of potential sensitivities regarding data from ethnic minorities. We have included contact information for the ethics committee via in fijihealthresearch@gmail.com. We recommend that requests for data also be sent to the Principal Investigator, Prof. Fiona Russell (fmruss@unimelb. edu.au), so she can assist with the process.

**Funding:** This project was funded by the Bill & Melinda Gates Foundation (grant numbers OPP1126272 and OPP1084341), and the Department of Foreign Affairs and Trade of the Australian Government, the Fiji Health Sector Support Program (implemented by Abt JTA on behalf of the Australian Government), with support from the Victorian Government's Operational Infrastructure Support Program (https://www2. health.vic.gov.au/about/clinical-trials-and-research/ operational-infrastructure-support). FMR was supported by a NHMRC Early Career and TRIP Fellowships (https://www.nhmrc.gov.au/). CS was supported by a NHMRC Career Development Fellowship (1087957) and a veski Inspiring Women Fellowship (https://www.veski.org.au/). EFGN holds an Australian Government Research Training Scholarship (https://www.education.gov. au/research-training-program). The funders had no role in study design, data collection and analysis, decision to publish, or preparation of the manuscript.

**Competing interests:** I have read the journal's policy and the authors of this manuscript have the following competing interests: CDN, CS, EKM, and EMD are investigators on a research projected funded by Pfizer for research in Mongolia. This does not alter our adherence to PLOS ONE policies on sharing data and materials. All other authors have no declarations of competing interests to declare

associated with pneumococcal carriage compared with Fijians of Indian Descent, despite similar PCV10 coverage rates.

## Introduction

Pneumococcal disease is a leading cause of childhood morbidity and mortality worldwide [1]. Pneumococcal disease is preceded by pneumococcal nasopharyngeal carriage, and the severity of pneumococcal pneumonia is associated with bacterial load (density) of pneumococci in the nasopharynx [2, 3]. Public health interventions to prevent pneumococcal disease can be improved by identifying the factors associated with pneumococcal carriage. Determining factors associated with higher pneumococcal carriage density could aid estimation of pneumococcal pneumonia prevalence in childhood pneumonia studies [2].

In low- and middle-income countries, risk factors for pneumococcal nasopharyngeal carriage vary [4–8] and few studies have investigated factors associated with the density of pneumococcal carriage in healthy populations [2, 9–11]. Common factors positively associated with pneumococcal carriage in low- and middle-income countries include indigenous ethnicity, passive smoking; co-colonisation with *Haemophilus influenzae*, childcare attendance, poverty, acute respiratory infection, living with young children, and being under five years old [4, 8, 12]. In studies from low- and middle-income countries, higher pneumococcal density has been positively associated with the symptoms of upper respiratory tract infection, presence of a febrile acute respiratory infection in children [13], rainy season, severe pneumonia, viral co-infection, radiologically confirmed pneumococcal pneumonia, and encapsulated serotypes (compared with non-encapsulated serotypes) [2, 9–11, 13–16].

Pneumococcal conjugate vaccines (PCV) reduce vaccine-type carriage and disease [17, 18]. However, serotype replacement with non-vaccine-type carriage occurs following the introduction of PCV which can result in non-vaccine-type disease [19, 20]. In the post-PCV era, it is largely unknown what the impact of PCV is on the risk factors for pneumococcal carriage and density in low- and middle-income countries in the Asia-Pacific region.

In 2012, Fiji introduced the ten-valent PCV (PCV10). Six years before PCV10 was introduced, factors associated with pneumococcal carriage in healthy 3–13 month old Fijians included indigenous iTaukei (iTaukei) ethnicity and having symptoms of an upper respiratory tract infection (URTI)[4]. iTaukei ethnicity was also associated with higher median pneumococcal density among 17 month old Fijians [21]. Fiji provides an opportunity to investigate factors associated with pneumococcal carriage and density in the post-PCV10, in a tropical, upper middle-income setting. As part of a PCV10 impact evaluation on pneumococcal carriage, we previously reported that the prevalence of vaccine-type carriage declined in both iTaukei and Fijians of Indian Descent (FID) three years after the introduction of PCV10, but carriage of non-vaccine-type carriage increased in iTaukei infants and toddlers [22]. The aim of this study is to determine the factors associated with carriage and density of pneumococci (overall, vaccine-type, and non-vaccine-type) in Fiji up to three years following PCV10 introduction.

## Materials and methods

### Setting

The majority of the Fijian population (81%) lives on Viti Levu, the bigger of Fiji's two main islands. Greater than half (56.8%) of the population are iTaukei and 37.8% are FID. This study

was conducted in Suva, and the surrounding areas, where over one-third of the population lives [23]. PCV10 was introduced nationally in October 2012 to be administered at 6, 10, and 14 weeks of age, with no catch-up campaign. The national coverage of the third dose of PCV10 one, two, and three years post-introduction was 84.9%, 84.9%, and 89.0%, respectively [24–26].

## Cross-sectional carriage studies

The design and methods for these cross-sectional studies have been described elsewhere [22]. Briefly, four annual cross-sectional carriage surveys were undertaken: one pre-PCV10 (2012), and then annually thereafter (2013–2015). Purposive quota sampling achieved a sample proportionate to the national iTaukei: FID (3:2) and rural: urban (1:1) ratios [4, 23, 27]. Each year, approximately 500 participants were recruited into each of the following four groups: infants 5–8 weeks (infants), toddlers 12–23 months (toddlers), children 2–6 years (children), and their parents/guardians (caregivers). Age groups were selected in order to best answer the primary research question regarding impact of PCV10 on pneumococcal carriage prevalence in Fiji, and as described previously, were based on those most likely to benefit from direct and indirect effects of PCV10; those likely to have the highest pneumococcal carriage prevalence; those age-eligible for PCV10 vaccination; and those most likely to transmit or be in contact with transmitters of pneumococci [22]. This analysis used the same study population.

Participants were recruited from two health centres in the Greater Suva area, and from surrounding communities. Eligibility criteria included age or being a caregiver, and for non-infant participants, that they had lived in the area for at least three months. Those with an axillary temperature ≥37.0°C were excluded. For the pre-PCV10 survey, any receipt of PCV10 was an exclusion criterion for all participant groups. For subsequent surveys, only infants who had ever received PCV10 were excluded.

Study staff interviewed caregivers and recorded individual level participant characteristics on data collection forms. Variables collected included: self-reported ethnicity, sex, residential location, antibiotic use in the fortnight preceding the survey, exposure to household cigarette smoke, coryza, symptoms of allergic rhinitis, cough, ear discharge, number of children less than five years in the household, and weekly family income. Caregivers reported their own, and their participating child's or children's ethnicity, according to options defined by the investigator (iTaukei, FID, or other) and recognized by Fijian population[23]. PCV10 vaccination status for infants and children was obtained from written records. As PCV10 was unavailable privately, caregivers were assumed to be PCV10 unvaccinated. A binary variable for symptoms of URTI was derived from the presence of one or more of the following: coryza, allergic rhinitis, cough, or ear discharge. A binary variable for low family income was derived, defined as family income on / above or below the basic needs poverty line (<FJ$175 per week) [28].

Trained study nurses collected nasopharyngeal samples using flocked nylon swabs (COPAN FLOQSwabs$^{TM}$), which were transported and stored according to standard methods, as described previously [22, 29]. Microbiological analyses were undertaken at the Murdoch Children's Research Institute in Melbourne, Australia as described previously [22]. In brief, pneumococci were detected using real-time quantitative-polymerase chain reaction targeting the *lytA* gene, with molecular serotyping by microarray [29–31]. Laboratory staff were blinded to participant characteristic data. Detection of any pneumococci in swab samples, including non-encapsulated lineages, was defined as overall pneumococcal carriage. Detection of serotypes included in PCV10 (serotypes 1, 4, 5, 6B, 7F, 9V, 14, 18C, 19F, and 23F) was defined as PCV10 carriage, and detection of serotypes not included in PCV10, including non-

   

**Table 1. Characteristics of participants in four annual cross-sectional community nasopharyngeal carriage surveys, 2012–2015, Fiji (n = 8,109 [a]).**

| Characteristics | | Summary statistic |
|---|---|---|
| **PCV10 vaccinated[b], n (%)** | | 1105 (13.6) |
| **Survey year, n (%)** | | |
| | Pre-PCV10 (2012) | 2025 (25.0) |
| | 1 year post-PCV10 (2013) | 2042 (25.2) |
| | 2 years post-PCV10 (2014) | 2022 (24.9) |
| | 3 years post PCV10 (2015) | 2020 (24.9) |
| **Ethnicity, n (%)** | | |
| | Fijian of Indian Descent | 3236 (39.9) |
| | iTaukei | 4835 (59.6) |
| | Other | 38 (0.5) |
| **Participant group, n (%)** | | |
| | Infants (5–8 weeks) | 2006 (24.7) |
| | Toddlers (12–23 months) | 2004 (24.7) |
| | Children (2–6 years) | 2052 (25.3) |
| | Caregivers | 2047 (25.3) |
| **Residential location, n (%)** | | |
| | Rural | 3944 (48.6) |
| | Urban | 4165 (51.4) |
| **Female sex, n (%)** | | 4683 (57.8) |
| **Two or more children under five years in the household, n (%)** | | n = 8106 |
| | | 4004 (49.4) |
| **Low family income[c], n (%)** | | n = 7831 |
| | | 4599 (58.7) |
| **Symptoms of URTI, n (%)** | | 2092 (25.8) |
| **Exposure to household cigarette smoke, n (%)** | | 4353 (53.7) |
| **Antibiotic use in past fortnight, n (%)** | | n = 8105 |
| | | 357 (4.4) |
| **Pneumococcal carriage, n / N (%)** | | |
| | Overall[d] | n = 8061 |
| | | 2456 (30.5) |
| | PCV10 serotypes[e] | n = 8000 |
| | | 713 (8.9) |
| | Non-PCV10 serotypes[f] | n = 8000 |
| | | 1915 (23.9) |
| | Non-encapsulated pneumococci[g] | n = 8000 |
| | | 390 (4.9) |
| **Pneumococcal density[h], n, median $\log_{10}$ GE/ml (IQR)** | | |
| | Overall | 2455, 5.0 (4.2–5.7) |
| | PCV10 serotypes | 713, 4.9 (4.1–5.6) |
| | Non-PCV10 serotypes | 1915, 4.9 (4.1–5.7) |
| | Non-encapsulated pneumococci | 390, 4.3 (3.7–4.9) |

Abbreviations: PCV10, ten-valent pneumococcal conjugate vaccine; URTI, upper respiratory tract infection; IQR, interquartile range.

[a] Unless otherwise specified

[b] Two doses of PCV10 given before the age of 12 months, or one or more doses of PCV10 given at or after 12 months of age[35]

[c] Weekly family income below the basic needs poverty line (<FJ$175 per week)[28]

[d] Any pneumococci, including non-encapsulated lineages

[e] Pneumococcal serotypes included in PCV10 (serotypes 1, 4, 5, 6B, 7F, 9V, 14, 18C, 19F, and 23F)

[f] Pneumococcal serotypes not included in PCV10, including non-encapsulated lineages

[g] Includes carriage of any of the following non-encapsulated lineages: NT, NT1, NT2, NT2/NT3b, NT3a, NT3b, NT4a, NT4b

[h] Only includes participants who were carriers of indicated pneumococcal types

encapsulated lineages, was defined as non-PCV10 carriage. Detection by microarray of a PCV10 serotype and a non-PCV10 serotype from the same swab sample was recorded as positive for both PCV10 and non-PCV10 serotype carriage [22]. Any detection of a serotype by microarray was considered positive, regardless of relative abundance [22]. Non-encapsulated lineages were classified based upon previously described genetic variants [32]. We determined pneumococcal density only for pneumococcal positive samples, and reported it in genome equivalents per ml (GE / ml). Participant characteristic data were double entered, and validated, in an EpiData 3.1 database [33]. Microbiological outcome data were entered into Microsoft Excel (Excel 2013) and merged with characteristic data in Stata 15.1 [34].

## Statistical analyses

Participant characteristics were summarised using counts and percentages. We built logistic and quantile regression models to investigate the factors associated with pneumococcal carriage and density, respectively. Empirical univariable results (p<0.2) and *a priori* selection informed variable choice for multivariable models. Factors assessed empirically included residential location, participant sex, two or more children under five years living in the household, low family income, exposure to household cigarette smoke, and recent antibiotic use. Variables selected *a priori* included PCV10 vaccination, survey year, ethnicity, participant group, and URTI symptoms [4, 14]. Interaction terms for ethnicity with PCV10 vaccination status, and with survey year, were assessed to account for the potential differential effect of PCV10 vaccination, or number of years post-PCV10 introduction, on pneumococcal carriage and density by ethnicity. We also assessed potential interaction between ethnicity and other socio-demographic factors in the models. Interaction terms were included as indicated, with significance level set at $P < 0.05$. Estimates of the association of participant characteristics with carriage and density were reported as odds ratios and differences in medians, respectively, with 95% confidence intervals (95% CI) and $P$-values. Pneumococcal density data were $\log_{10}$ transformed prior to analyses, and analyses of pneumococcal density were restricted to pneumococcal carriers. Only 14/38 participants who identified as "other" ethnicity had pneumococcal positive samples, so were excluded from inferential analyses. Merged data were cleaned and analysed in Stata 15.1 [34].

## Ethics statement

This study was carried out in accordance with the protocols approved by the Fijian National Health Research and Ethics Review Committee (201228), and The University of Melbourne Health Sciences Human Ethics Sub-Committee (1238212). Study staff discussed the study with caregivers, and written informed consent was completed prior to any study procedures. Participants were not offered any incentive to participate.

## Results

### Participant characteristics

There were 8,109 participants, with characteristics shown in Table 1. The overall vaccination rate of 13.6% reflects the pooled participant group, most (85.8%) of which were not age-eligible to receive PCV10. Similar numbers of people participated per survey year and by participant group [22]. Few had used antibiotics in the preceding two weeks. Forty-eight participant swab samples were excluded from microbiological analysis due to insufficient volume, sample loss, or labelling errors. A further 61 pneumococcal positive samples were excluded from serotyping due to biological reasons or technical issues. Of the 8,061 participants for whom swab sample

results were available, 30.5% tested positive for pneumococci. Among the 8,000 serotyped samples, PCV10 carriage was uncommon (8.9%), and 23.9% of participants carried non-PCV10 pneumococci. Carriage of non-encapsulated pneumococci was rare (390 / 8,000, 4.9%). Density data was unavailable for one pneumococcal carrier. Overall carriage median density was 5.0 $\log_{10}$ GE/ml (4.2–5.7), while those for PCV10, non-PCV10 carriage, and non-encapsulated lineages were 4.9 $\log_{10}$ GE/ml (4.1–5.6), 4.9 $\log_{10}$ GE/ml (4.1–5.7), and 4.3 $\log_{10}$ GE/ml (3.7–4.9), respectively [22].

## Factors associated with overall carriage

ITaukei ethnicity, young age (infant, toddler, and child participant groups vs caregivers), urban residence, living with two or more children under five years, low family income, and URTI symptoms were positively associated with overall carriage (Table 2). Survey year was negatively associated with overall carriage. There was evidence of protection from PCV10 vaccination against overall carriage, however the confidence interval crossed the null value. There was evidence of an interaction between survey year and ethnicity (global $P<0.001$), but no evidence of an interaction between PCV10 vaccination status and ethnicity (global $P = 0.880$).

## Factors associated with PCV10 carriage

iTaukei ethnicity, young age (infant, toddler, and child participant groups), urban residence, living with two or more children under five years, low family income, symptoms of URTI, and exposure to household cigarette smoke were positively associated with PCV10 carriage (Table 3). PCV10 vaccination and survey year were negatively associated with PCV10 carriage. There was no evidence of an interaction between PCV10 vaccination status ($P = 0.902$) or survey year ($P = 0.171$) and ethnicity with regard to PCV10 carriage.

## Factors associated with non-PCV10 carriage

iTaukei ethnicity, young age (infant, toddler, and child participant groups), urban residence, living with two or more children younger than five years, low family income, and URTI symptoms were positively associated with non-PCV10 carriage (S1 Table). As with overall carriage, there was evidence of an interaction between survey year and ethnicity, as the two ethnic groups had differential odds of non-PCV10 carriage ($P<0.001$), but no evidence of interaction between PCV10 vaccination status and ethnicity ($P = 0.856$).

## Factors associated with overall pneumococcal density

Toddler and child participant groups, and symptoms of URTI were positively associated with density of overall pneumococcal carriage (Table 4). There was evidence of an association between iTaukei ethnicity and overall pneumococcal density, however the confidence interval included the null value. Although the adjusted median difference in overall pneumococcal carriage density increased in the first two years after the introduction of PCV10, this was not sustained into the third year (Table 4). There was no indication of an interaction between PCV10 vaccination status ($P = 0.864$) or survey year ($P = 0.347$) with ethnicity.

## Factors associated with PCV10 pneumococcal density

Symptoms of URTI were positively associated with median density of PCV10 carriage (S2 Table). Although there was an initial increase in the adjusted median difference in PCV10 carriage density in the first year after PCV10 introduction, it was not sustained through the second and third year post-PCV10 introduction (S2 Table). There was no indication of an

**Table 2. Unadjusted and adjusted odds ratios of overall pneumococcal carriage in association with participant characteristics in four cross-sectional carriage surveys pre-PCV10 (2012) and annually thereafter (2013–2015) in Fiji (n = 8,023).**

| Exposure | Overall carriage[a] n/N (%) | Unadjusted odds ratio (95% CI) | *P*-value | Adjusted odds ratio (95% CI) | *P*-value |
|---|---|---|---|---|---|
| **PCV10 vaccination status** | | | <0.001 | | 0.065 |
| Not vaccinated | 2027 / 6931 (29.3) | *ref* | | *ref* | |
| Vaccinated[b] | 415 / 1092 (38.0) | 1.48 (1.30–1.69) | | 0.82 (0.66–1.01) | |
| **Survey year** | | | <0.001 | | <0.001 |
| Pre-PCV10 (2012) | 708 / 2001 (35.4) | *ref* | | *ref* | |
| 1 year post-PCV10 (2013) | 655 / 2033 (32.2) | 0.87 (0.76–0.99) | | 0.67 (0.51–0.88) | |
| 2 years post-PCV10 (2014) | 433 / 1997 (21.7) | 0.51 (0.44–0.58) | | 0.49 (0.36–0.66) | |
| 3 years post PCV10 (2015) | 646 / 1992 (32.4) | 0.88 (0.77–1.00) | | 0.62 (0.46–0.83) | |
| **Ethnicity** | | | <0.001 | | <0.001 |
| Fijian of Indian Descent | 496 / 3218 (15.4) | *ref* | | *ref* | |
| iTaukei | 1946 / 4805 (40.5) | 3.74 (3.34–4.18) | | 2.74 (2.17–3.45) | |
| **Participant group** | | | <0.001 | | <0.001 |
| Caregivers | 193 / 2035 (9.5) | *ref* | | *Ref* | |
| Infants (5–8 weeks) | 516 / 1974 (26.1) | 3.38 (2.82–4.04) | | 4.15 (3.40–5.06) | |
| Toddlers (12–23 months) | 845 / 1986 (42.6) | 7.07 (5.95–8.40) | | 8.88 (7.13–11.07) | |
| Children (2–6 years) | 888 / 2028 (43.8) | 7.43 (6.26–8.83) | | 8.48 (6.99–10.29) | |
| **Residential location** | | | <0.001 | | <0.001 |
| Rural | 1070 / 3911 (27.4) | *ref* | | *ref* | |
| Urban | 1372 / 4112 (33.4) | 1.33 (1.21–1.46) | | 1.45 (1.30–2.57) | |
| **Participant sex** | | | <0.001 | | 0.300 |
| Male | 1167 / 3385 (34.5) | *ref* | | *ref* | |
| Female | 1275 / 4638 (27.5) | 0.72 (0.65–0.79) | | 1.06 (0.95–1.19) | |
| **Number of children < 5 years living in the household[c]** | | | <0.001 | | <0.001 |
| Less than two | 955/4067 (23.5) | *ref* | | *ref* | |
| Two or more | 1487 / 3953 (37.6) | 1.96 (1.78–2.16) | | 1.42 (1.27–1.59) | |
| **Family income level[d]** | | | <0.001 | | <0.001 |
| Not low | 801 / 3193 (25.1) | *ref* | | *ref* | |
| Low | 1523 / 4558 (33.4) | 1.50 (1.35–1.66) | | 1.44 (1.28–1.62) | |
| **Symptoms of URTI** | | | <0.001 | | <0.001 |
| Not present | 1529 / 5955 (25.7) | *ref* | | *ref* | |
| Present | 913 / 2068 (44.2) | 2.29 (2.06–2.54) | | 1.77 (1.57–2.01) | |
| **Household cigarette smoke** | | | 0.105 | | 0.555 |
| No exposure | 1099 / 3729 (29.5) | *ref* | | *ref* | |
| Exposure | 1343 / 4303 (31.2) | 1.08 (0.98–1.19) | | 0.97 (0.87–1.08) | |
| **Antibiotic use in previous fortnight[e]** | | | 0.350 | | |
| Not used | 2325 / 7667 (30.3) | *ref* | | | |
| Used | 115 / 352 (32.7) | 1.11 (0.89–1.40) | | | |

Abbreviations: CI, confidence interval; PCV10, ten-valent pneumococcal conjugate vaccine; URTI, upper respiratory tract infection.

[a] Any pneumococci, including non-encapsulated lineages

[b] Two doses of PCV10 given before the age of 12 months, or one or more doses of PCV10 given at or after 12 months of age[35]

[c] Data on number of children under five years living in the household were missing for three participants, of whom none were pneumococcal carriers

[d] Weekly family income below the basic needs poverty line (<FJ$175 per week)[28]; data on family income were missing for 272 participants, of whom 118 were pneumococcal carriers

[e] Data on antibiotics use were missing for four participants, of whom two were pneumococcal carriers.

**Table 3. Unadjusted and adjusted odds ratios of PCV10 pneumococcal carriage in association with participant characteristics in four cross-sectional carriage surveys pre-PCV10 (2012) and annually thereafter (2013–2015) in Fiji (n = 7,962).**

| Exposure | PCV10 carriage[a] n / N (%) | Unadjusted odds ratio (95% CI) | P-value | Adjusted odds ratio (95% CI) | P-value |
|---|---|---|---|---|---|
| **PCV10 vaccination status** | | | 0.043 | | 0.002 |
| Not vaccinated | 629 / 6875 (9.2) | ref | | ref | |
| Vaccinated[b] | 79 / 1087 (7.3) | 0.78 (0.61–0.99) | | 0.58 (0.41–0.82) | |
| **Survey year** | | | <0.001 | | <0.001 |
| Pre-PCV10 (2012) | 275 / 1975 (13.9) | ref | | ref | |
| 1 year post-PCV10 (2013) | 216 / 2022 (10.7) | 0.74 (0.61–0.89) | | 0.74 (0.60–0.91) | |
| 2 years post-PCV10 (2014) | 102 / 1987 (5.13) | 0.33 (0.26–0.42) | | 0.40 (0.30–0.53) | |
| 3 years post PCV10 (2015) | 115 / 1978 (5.8) | 0.38 (0.30–0.48) | | 0.46 (0.35–0.61) | |
| **Ethnicity** | | | <0.001 | | <0.001 |
| Fijian of Indian Descent | 147 / 3206 (4.6) | ref | | ref | |
| iTaukei | 561 / 4756 (11.8) | 2.78 (2.31–3.36) | | 2.70 (2.21–3.30) | |
| **Participant group** | | | <0.001 | | <0.001 |
| Caregivers | 41 / 2029 (2.0) | ref | | ref | |
| Infants (5–8 weeks) | 121 / 1946 (6.2) | 3.21 (2.24–4.61) | | 3.60 (2.45–5.30) | |
| Toddlers (12–23 months) | 277 / 1972 (14.1) | 7.92 (5.67–11.07) | | 9.76 (6.67–14.37) | |
| Children (2–6 years) | 269 / 2015 (13.4) | 7.47 (5.34–10.44) | | 7.65 (5.32–11.00) | |
| **Residential location** | | | 0.001 | | 0.001 |
| Rural | 304 / 3880 (7.8) | ref | | ref | |
| Urban | 404 / 4082 (9.9) | 1.29 (1.11–1.51) | | 1.34 (1.13–1.58) | |
| **Participant sex** | | | <0.001 | | 0.772 |
| Male | 352 / 3358 (10.5) | ref | | ref | |
| Female | 356 / 4604 (7.7) | 0.72 (0.61–0.84) | | 0.98 (0.82–1.16) | |
| **Number of children < 5 years living in the household[c]** | | | <0.001 | | 0.040 |
| Less than two | 283 / 4035 (7.0) | ref | | ref | |
| Two or more | 425 / 3924 (10.8) | 1.61 (1.38–1.88) | | 1.20 (1.01–1.43) | |
| **Family income level[d]** | | | <0.001 | | 0.004 |
| Not low | 207 / 3175 (6.5) | ref | | ref | |
| Low | 463 / 4518 (10.3) | 1.64 (1.38–1.94) | | 1.31 (1.09–1.57) | |
| **Symptoms of URTI** | | | <0.001 | | <0.001 |
| Not present | 435 / 5903 (7.4) | ref | | ref | |
| Present | 273 / 2059 (13.3) | 1.92 (1.64–2.26) | | 1.42 (1.19–1.70) | |
| **Household cigarette smoke** | | | <0.001 | | 0.031 |
| No exposure | 283 / 3689 (7.7) | ref | | ref | |
| Exposure | 425 / 4273 (10.0) | 1.33 (1.14–1.56) | | 1.21 (1.02–1.43) | |
| **Antibiotic use in previous fortnight[e]** | | | 0.012 | | 0.367 |
| Not used | 663 / 7610 (8.7) | ref | | ref | |
| Used | 44 / 348 (12.6) | 1.52 (1.09–2.10) | | 0.84 (0.58–1.22) | |

Abbreviations: CI, confidence interval; PCV10, ten-valent pneumococcal conjugate vaccine; URTI, upper respiratory tract infection.

a Pneumococcal serotypes included in PCV10 (serotypes 1, 4, 5, 6B, 7F, 9V, 14, 18C, 19F, and 23F)

b Two doses of PCV10 given before the age of 12 months, or one or more doses of PCV10 given at or after 12 months of age [35]

c Data on number of children under five years living in the household were missing for three participants, of whom none were PCV10 pneumococcal carriers

d Weekly family income below the basic needs poverty line (<FJ$175 per week) [28]; data on family income were missing for 269 participants, of whom 38 were PCV10 pneumococcal carriers

e Data on antibiotics use were missing for four participants, of whom one was a PCV10 pneumococcal carrier.

interaction between PCV10 vaccination status ($P$ = 0.170) or survey year ($P$ = 0.686) with ethnicity.

### Factors associated with non-PCV10 pneumococcal density

Symptoms of an URTI and young age (infant, toddler, and child participant groups) were positively associated with median density of non-PCV10 carriage (S3 Table). As with density of overall and PCV10 pneumococcal carriage, the increase in density of non-PCV10 carriage in association with survey year was not sustained to the third year post-PCV10 introduction, and there was no evidence of interaction between PCV10 vaccination status ($P$ = 0.156) survey year ($P$ = 0.138) with ethnicity.

## Discussion

This study provides new findings on the factors associated with overall, PCV10, and non-PCV10 pneumococcal carriage and density, in the years surrounding PCV10 introduction in a tropical, upper middle-income country in the Asia-Pacific region.

In this study, iTaukei ethnicity was an independent predictor of carriage (overall, PCV10, and non-PCV10), after adjustment for PCV10 vaccination status and survey year post-PCV10 introduction. We also checked the potential interaction between ethnicity and other socio-demographic variables, and found no evidence of interaction. Previously, we found that prior to PCV10 introduction, iTaukei ethnicity was associated with increased odds of overall pneumococcal carriage in children aged 3–13 months (aOR 2.81 [95% CI 1.76–4.49] $P$<0.001) [4]. The reason for this association is unknown. Further, we have reported that differences in social contact patterns by ethnicity partially account for higher prevalence of pneumococci among iTaukei, compared with FID, but that differences in carriage prevalence are also likely related to ethnic differences in host or environmental factors [36]. Few studies have described associations between pneumococcal carriage and indigenous and non-indigenous populations living in the same area, with similar access to healthcare, and with similar and high PCV coverage rates. Our findings are consistent with other studies comparing indigenous and non-indigenous populations in the same setting. In a pre- and very early post-PCV7 longitudinal study of 280 indigenous and non-indigenous children in remote Australia, who were followed from birth to 2 years, indigenous status was positively associated with pneumococcal carriage (OR 3.3 [95% CI 2.6–4.2] $P$<0.001)[37]. In Israel, a longitudinal study of 369 Bedouin and 400 Jewish children enrolled in a trial of PCV7, found Jewish children to have significantly lower odds of pneumococcal carriage, compared with Bedouin children (aOR 0.14 [0.10–0.21]) [38]. In contrast, a post-PCV13 cross-sectional study of 352 children aged less than six years in Greenland, found indigenous ethnicity was not associated with pneumococcal carriage (aOR 0.7 [95% CI 0.3–1.5] $P$ = 0.32) [39]. Likewise, a cross-sectional study in Alaska post-PCV7 involving 1,275 children aged 3–59 months, found no association between indigenous ethnicity and overall or PCV7 carriage (OR 1.0 [95% CI 0.8–1.3] and (OR 1.1 [95% CI 0.75–1.6], respectively) [40]. Unlike our study, however, both the Greenland and Alaska studies occurred in high income settings, which may comparatively reduce the impact of ethnicity based socio-environmental differences that might be related to pneumococcal carriage. Further, the Greenland study included few non-Inuit participants, and may have been underpowered for analysis by ethnicity [39].

Other factors associated with pneumococcal carriage in this study, were largely consistent with studies from the pre-PCV10 era. For example, we observed young age, residential location, living with young children, low family income, and symptoms of URTI to be risk factors for all types of carriage [4, 8, 12]. Similarly, the majority of studies assessing factors associated

**Table 4. Unadjusted and adjusted differences in medians of overall pneumococcal carriage density in association with participant characteristics in four cross-sectional carriage surveys pre-PCV10 (2012) and annually thereafter (2013–2015) in Fiji (n = 2,441).**

| Exposure | Density of overall pneumococcal carriage[a] (log10 GE/ml) n, median / IQR | Unadjusted mean difference (95% CI) | P-value | Adjusted mean difference (95% CI) | P-value |
|---|---|---|---|---|---|
| **PCV10 vaccination status** | | | 0.408 | | 0.210 |
| Not vaccinated | 2026, 4.9 (4.2–5.8) | ref | | Ref | |
| Vaccinated[b] | 415, 4.9 (4.0–5.7) | -0.06 (-0.21, 0.08) | | -0.13 (-0.35, 0.08) | |
| **Survey year** | | | 0.004 | | 0.001 |
| Pre-PCV10 (2012) [b] | 707, 4.9 (4.2–5.7) | ref | | ref | |
| 1 year post-PCV10 (2013) | 655, 5.2 (4.3–5.9) | 0.28 (0.13, 0.43) | | 0.33 (0.18, 0.47) | |
| 2 years post-PCV10 (2014) | 433, 5.0 (4.1–5.8) | 0.11 (-0.06, 0.27) | | 0.18 (0.00, 0.36) | |
| 3 years post-PCV10 (2015) | 646, 4.9 (4.0–5.7) | 0.00 (-0.15, 0.15) | | 0.01 (-0.17, 0.18) | |
| **Ethnicity** | | | 0.012 | | 0.053 |
| Fijian of Indian Descent | 495, 4.8 (4.1–5.7) | ref | | ref | |
| iTaukei | 1946 5.0 (4.2–5.8) | 0.17 (0.04, 0.31) | | 0.14 (0.00, 0.28) | |
| **Participant group** | | | 0.003 | | 0.008 |
| Caregivers | 193, 4.7 (4.0–5.4) | ref | | ref | |
| Infants (5–8 weeks) | 515, 4.9 (4.1–5.7) | 0.17 (-0.06, 0.39) | | 0.17 (-0.06, 0.40) | |
| Toddlers (12–23 months) | 845, 5.0 (4.1–5.8) | 0.33 (0.11, 0.55) | | 0.32 (0.08, 0.55) | |
| Children (2–6 years) | 888, 5.0 (4.3–5.8) | 0.34 (0.12, 0.55) | | 0.34 (0.12, 0.55) | |
| **Residential location** | | | 0.612 | | |
| Rural | 1070, 5.0 (4.1–5.7) | ref | | | |
| Urban | 1371, 5.0 (4.2–5.7) | 0.03 (-0.08, 0.14) | | | |
| **Participant sex** | | | 0.597 | | |
| Male | 1166, 5.0 (4.2–5.7) | ref | | | |
| Female | 1275, 5.0 (4.2–5.8) | 0.03 (-0.08, 0.14) | | | |
| **Number of children < 5 years living in the household** | | | 0.708 | | |
| Less than two | 954, 5.0 (4.2–5.8) | ref | | | |
| Two or more | 1487, 5.0 (4.2–5.7) | -0.02 (-0.13, 0.09) | | | |
| **Family income level[c]** | | | 0.943 | | |
| Not low | 801, 5.0 (4.1–5.7) | ref | | | |
| Low | 1522, 5.0 (4.2–5.8) | 0.00 (-0.11, 0.12) | | | |
| **Symptoms of URTI** | | | <0.001 | | <0.001 |
| Not present | 1528, 4.9 (4.1–5.6) | ref | | Ref | |
| Present | 913, 5.2 (4.3–5.9) | 0.31 (0.19, 0.42) | | 0.28 (0.16, 0.40) | |
| **Household cigarette smoke** | | | 0.540 | | |
| No exposure | 1098, 5.0 (4.1–5.7) | ref | | | |
| Exposure | 1343, 5.0 (4.2–5.8) | 0.03 (-0.08, 0.15) | | | |
| **Antibiotic use in previous fortnight[d]** | | | 0..084 | | 0.752 |
| Not used | 2324, 5.0 (4.2–5.7) | ref | | ref | |
| Used | 115, 5.2 (4.5–5.7) | 0.22 (-0.03, 0.48) | | 0.04 (-0.23, 0.30) | |

Abbreviations: CI, confidence interval; PCV10, ten-valent pneumococcal conjugate vaccine; URTI, upper respiratory tract infection.

[a] Density of overall, including non-encapsulated, pneumococci

[b] Two doses of PCV10 given before the age of 12 months, or one or more doses of PCV10 given after 12 months of age [35]

[c] Weekly family income below the basic needs poverty line (<FJ$175 per week) [28]; data on family income were missing for 118 pneumococcal carriers

[d] Data on antibiotic use were missing for two pneumococcal carriers.

with pneumococcal carriage after the introduction of PCV into national immunization schedules have reported age, poverty or proxies of poverty, number of young children living in the household, and symptoms of URTI as positively associated with pneumococcal carriage [41–45]. Previous studies have heterogenous findings regarding exposure to cigarette smoke and recent antibiotic use and their associations with pneumococcal carriage [5, 11, 46–53]. We found exposure to household cigarette smoke was a risk factor, but only for PCV10 carriage. However, levels of exposure to cigarette smoke require detailed monitoring, which was not incorporated in this study. We also found no association between antibiotic use and pneumococcal carriage, which may reflect very low prevalence of recent antibiotic use in our sample.

In this study, PCV10 vaccination status and survey year were protective against overall and PCV10 carriage, but were not associated with non-PCV10 carriage. We found only three studies undertaken after PCV was introduced into national immunization programs that assessed factors associated with pneumococcal carriage, and included PCV vaccination status as a variable [16, 54, 55]. In a cross-sectional study of 361 children under five years of age in Jimma, Ethiopia, the odds of overall pneumococcal carriage increased in association with having siblings under five years old (aOR 1.80 [95% CI 1.17–2.77] P = 0.008), and malnutrition (aOR 2.07 [95% CI 1.24–3.44] P = 0.005), but PCV vaccination was not associated with a decrease in carriage (three doses aOR 1.08 [95% CI 0.60–1.89] P = 0.82, one or two doses aOR 1.06 [95% CI 0.40–2.83] P = 0.90) [54]. This may have been due to serotype replacement and capsular switching of pneumococci by recombination, such that the immune pressure from PCV selected for non-vaccine serotypes [54]. However, in a cross-sectional study of 1,668 healthy children aged 12–23 months in Brazil, the odds of vaccine-type carriage declined in association with three (aOR 0.073 [95% CI 0.026–0.204] P<0.001) and four (aOR 0.027 [95% CI 0.007–0.113] P<0.001) doses of PCV10, and increased in association with day care attendance (aOR 2.358 [95% CI 1.455–3.821] P<0.001) and colonization with *H. influenzae* (aOR 2.454 [95% CI 1.529–3.939] P = 0.0006) [55]. Similarly, in pre and post-PCV13 pneumococcal carriage surveys involving 999 infants aged 5–8 weeks and 1,010 toddlers aged 12–23 months in Lao PDR, two or three doses of PCV13, compared with zero or one, was protective against PCV13 carriage among toddlers (aOR 0.60 [95% CI 0.44–0.83] P = 0.002) [16].

Although our findings are consistent with the Brazilian and Lao PDR studies regarding PCV being protective against PCV carriage, broader comparisons between these and other studies is hampered by the heterogeneity of settings, populations sampled, and the factors and definitions used. For example, in our study participants were community based, healthy, from four different age groups, did not attend childcare, and we did not include co-colonisers or malnutrition as exposures. Comparatively, the studies from Ethiopia, Brazil, and Lao PDR included child and infant participants only, including those attending childcare / school, and those suffering malnutrition, pneumonia, sinusitis, and otitis media.

Our observations regarding no association between PCV10 and non-PCV10 carriage may be due a lack of selection pressure towards an overall increase in non-PCV10 carriage early post-PCV10 introduction, due to serotype replacement occurring only in iTaukei infants and toddlers, rather than across all participant groups, as described previously [22]. In contrast, increases in non-vaccine type carriage have been reported following the introduction of PCV7 in England, PCV10 in Kenya, and PCV13 in Malawi and The Gambia [56–59]. However, it should be noted that the studies from Kenya, England, The Gambia, and Malawi were in vastly different contexts from our study, one notably in a high-income setting, rendering comparisons difficult.

Pneumococcal density has previously been found to be positively associated with microbiologically confirmed pneumococcal pneumonia, and could be used to improve estimates of pneumococcal pneumonia prevalence in childhood pneumonia studies [2].Our study

contributes to the understanding of factors associated with pneumococcal carriage density. We found that symptoms of an URTI were associated with increased median density of carriage (overall, PCV10, and non-PCV10), consistent with cross-sectional carriage surveys from Peru, Lao PDR, and Indonesia [9, 11, 13, 16]. We found PCV10 vaccination was not associated with pneumococcal density, and that although differences in median density of all types of pneumococci increased one to two years following PCV10 introduction, this was not sustained into the third year. There are relatively few risk factor studies describing the association between pneumococcal density and PCV vaccination status. Those that do, have heterogeneous findings. A double-blind, randomized controlled trial of PCV13 and Hepatitis A vaccine (control arm) in adults, using the Experimental Human Pneumococcal Challenge model, found that pneumococcal density in the PCV13 arm was significantly lower compared with the control arm ($P<0.0001$) [60]. In a cluster-randomized trial of PCV7 and Meningococcal C vaccine (control arm) in rural Gambia, density of PCV7 pneumococcal carriage was lower in PCV7 villages compared with control arm villages [61]. However, this was not attributed to PCV7, as density of non-PCV7 carriage also declined in both vaccine and control villages [61]. Similarly, a decline observed in both PCV13 and non-PCV13 pneumococcal density in Laotian infants and toddlers was attributed to secular trends rather than PCV13 directly [16].

Limitations to our study should be noted. Firstly, because participants were recruited from Greater Suva and the surrounding areas, generalizability to the wider Fijian population may be limited. The non-random sampling method may have introduced sample selection bias, such results may not be generalizable to the Fijian population. However, the purposive quota sampling method rendered the sample representative of the Fijian population with regard to ethnicity and residential location, which are associated with pneumococcal carriage [4, 27]. The cross-sectional nature of this observational study precludes causal, associations from being drawn between participant characteristics and pneumococcal carriage or densities. Finally, we did not collect data on co-colonizing bacterial or viral species, and are therefore unable to investigate the association of such factors with pneumococcal carriage or density, which have been identified as risk factors in other studies [9–11].

These limitations notwithstanding, this study documents the factors associated with pneumococcal carriage and density post-PCV10 introduction in an upper middle-income country. Introduction of PCV10 was negatively associated with the odds of overall and PCV10 pneumococcal carriage in Fiji. However, iTaukei ethnicity remains positively associated with pneumococcal carriage in Fiji, despite high and similar PCV10 coverage rates across iTaukei and FID populations. Further research is warranted regarding the factors underlying the observed ethnicity based differences in pneumococcal carriage, and whether the impact of PCV10 on pneumococcal disease incidence differs by ethnicity in Fiji.

## Supporting information

**S1 Table. Unadjusted and adjusted odds ratios of non-PCV10 pneumococcal carriage in association with participant characteristics in four cross-sectional carriage surveys pre-PCV10 (2012) and annually thereafter (2013–2015) in Fiji (n = 7,962).**
(DOCX)

**S2 Table. Unadjusted and adjusted differences in medians of PCV10 pneumococcal carriage density in association with participant characteristics in four cross-sectional carriage surveys pre-PCV10 (2012) and annually thereafter (2013–2015) in Fiji (n = 708).**
(DOCX)

**S3 Table. Unadjusted and adjusted differences in medians of non-PCV10 pneumococcal carriage density in association with participant characteristics in four cross-sectional carriage surveys pre-PCV10 (2012) and annually thereafter (2013–2015) in Fiji (n = 1,905).** (DOCX)

## Acknowledgments

We acknowledge the significant contributions made by the study participants and families in Fiji, the staff of the New Vaccine Evaluation Project in Fiji, and the staff at the Murdoch Children's Research Institute.

## Author Contributions

**Conceptualization:** Fiona M. Russell.

**Data curation:** Eleanor F. G. Neal, Felista T. Ratu, Eileen M. Dunne, Rita C. Reyburn, Catherine Satzke.

**Formal analysis:** Eleanor F. G. Neal.

**Funding acquisition:** Catherine Satzke, E. Kim Mulholland, Fiona M. Russell.

**Investigation:** Eleanor F. G. Neal, Felista T. Ratu, Eileen M. Dunne, Belinda D. Ortika, Laura K. Boelsen, Lisi Tikoduadua, Evelyn Tuivaga, Catherine Satzke.

**Methodology:** Eleanor F. G. Neal, Cattram D. Nguyen, Eric Rafai, E. Kim Mulholland, Fiona M. Russell.

**Project administration:** Eleanor F. G. Neal, Felista T. Ratu, Fiona M. Russell.

**Resources:** Felista T. Ratu, Eileen M. Dunne, Mike Kama, Joseph Kado, Catherine Satzke, Eric Rafai, E. Kim Mulholland, Fiona M. Russell.

**Software:** Eleanor F. G. Neal, Cattram D. Nguyen.

**Supervision:** Cattram D. Nguyen, Eileen M. Dunne, Rita C. Reyburn, Catherine Satzke, Fiona M. Russell.

**Validation:** Cattram D. Nguyen.

**Visualization:** Eleanor F. G. Neal.

**Writing – original draft:** Eleanor F. G. Neal.

**Writing – review & editing:** Eleanor F. G. Neal, Cattram D. Nguyen, Felista T. Ratu, Eileen M. Dunne, Mike Kama, Belinda D. Ortika, Laura K. Boelsen, Joseph Kado, Lisi Tikoduadua, Rachel Devi, Evelyn Tuivaga, Rita C. Reyburn, Catherine Satzke, Eric Rafai, E. Kim Mulholland, Fiona M. Russell.

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
