## [Decision Letter · Decision Letter 0]

27 Jan 2020

PONE-D-19-35241

Factors associated with pneumococcal carriage and density in children and adults in Fiji, using four cross-sectional surveys

PLOS ONE

Dear Ms Neal,

Thank you for submitting your manuscript to PLOS ONE. After careful consideration, we feel that it has merit but does not fully meet PLOS ONE’s publication criteria as it currently stands. Therefore, we invite you to submit a revised version of the manuscript that addresses the points raised during the review process.

Thank you for submitting this very interesting work. We invite you to submit a revised version of the manuscript that addresses all the points raised by the reviewers. 

We would appreciate receiving your revised manuscript by Mar 12 2020 11:59PM. To enhance the reproducibility of your results, we recommend that if applicable you deposit your laboratory protocols in protocols.io, where a protocol can be assigned its own identifier (DOI) such that it can be cited independently in the future. For instructions see: http://journals.plos.org/plosone/s/submission-guidelines#loc-laboratory-protocols

We look forward to receiving your revised manuscript.

Kind regards,

Jose Melo-Cristino, M.D., Ph.D.

Academic Editor

PLOS ONE

"I have read the journal's policy and the authors of this manuscript have the following competing interests: CDN, CS, EKM, and EMD are investigators on a research projected funded by Pfizer for research in Mongolia. All other authors have no declarations of competing interests to declare."

Please respond by return email with your amended Competing Interests Statement and we will change the online submission form on your behalf.

Reviewers' comments:

Reviewer's Responses to Questions

**Comments to the Author**

1. Is the manuscript technically sound, and do the data support the conclusions?

Reviewer #1: Yes

Reviewer #2: Yes

2. Has the statistical analysis been performed appropriately and rigorously? 

Reviewer #1: Yes

Reviewer #2: Yes

3. Have the authors made all data underlying the findings in their manuscript fully available?

Reviewer #1: Yes

Reviewer #2: Yes

4. Is the manuscript presented in an intelligible fashion and written in standard English?

Reviewer #1: Yes

Reviewer #2: Yes

5. Review Comments to the Author

Reviewer #1: Manuscript Number: PONE-D-19-35241

Title: Factors associated with pneumococcal carriage and density in children and adults in Fiji, using four cross-sectional surveys.

Excellent work describing the factors associated with pneumococcal nasopharyngeal carriage and density in children and caregivers in Fiji, pre and post-introduction of PCV10. This article is a continuation of the previously published article of the same authors, and reference 22 of the present one: Dunne EM, Satzke C, Ratu FT, Neal EF, Boelsen LK, Matanitobua S, et al. Impact of 10-valent pneumococcal conjugate vaccine introduction on pneumococcal carriage in Fiji: results from four annual cross sectional carriage surveys. Lancet Glob Health. 2018;6(12):PE1375-E85.

The article is well written, methodology is, in general, sound (see comment 16 on pneumococcal density) and support the results obtained.

I only have some minor comments:

General comments

1. Vaccinated subjects in each age group are different: from non-vaccinated to different percentages of vaccinated depending on the year of the survey. However data have not been stratified by age-groups and analysed globally, so that the direct effect of vaccination on carriage might be somehow “diluted”.

2. In many places of the article it is stated that “young age was positively associated with pneumococcal carriage”. However overall, PCV10 serotypes, and non-PCV10 serotypes carriage was lower in infants than in toddles and children. The authors should better specify to what age groups they are refereeing when talking of young age” as a general word.

3. In the tables there is a P-value that in some items (survey year and participant group) is given in total for the other 3 categories. Please, explain if this P-value was calculated comparing the reference value (caregivers) with the addition of the results of all the other groups or if the comparison of each group with the reference one was the P-value expressed.

Detailed comments

Abstract

4. Line 23. I suggest adding “in October 2012” after “…post-introduction of ten-valent pneumococcal conjugate vaccine (PCV10).”

5. May be the authors should differentiate in the abstract between factors positively or negatively associated with carriage, in order to be clearer for the reader. For instance, in line 26 it is stated that “Factors associated with pneumococcal carriage were years post-PCV10 introduction (global P<0.001) …” and in line 31: “…PCV10 carriage was also associated with PCV10 vaccination (0.58 [95% CI 0.41 – 0.82] P=0.002)”.

As PCV10 vaccination and survey year were negatively associated with PCV10 carriage (line 184) and survey year was negatively associated with overall carriage (line 165), but other factors were positively associated to carriage I recommend to write this clearly in the abstract to avoid possible misunderstandings.

6. Abstract and end of discussion. Although the study is mostly descriptive, it would be desirable some conclusions of the study, not just the report of the data obtained.

Material and methods

7. Children between 8 weeks and 12 months were not sampled, probably because they were in the vaccination period. However I would suggest an explanation of this decision in material and methods.

8. Line 141. Percentage of vaccinated subjects was different in each age-group (no vaccinated in infants and caregivers, 23% vaccinated children in 2015 and nearly 100% toddlers vaccinated in 2014-2015). In fact, 1105 (13.6%) participants had been vaccinated. Could these differences in the vaccination rates have biased the percentage of carriage in the different age-groups? The direct effect of vaccination in carriage could only be measured in 13.6% of participants, most of them in the 12-23 months age- group.

9. Line 147. In the previous publication authors detailed that “19.8% of pneumococcal-positive samples contained more than one serotype.” I suppose that some of these samples contained PCV10 and non-PCV10 serotypes. In this case, to which group (PCV10 carrier or not) were assigned the subjects?

10. Table 1. I suggest to put in this table and in the following, the year in the “Survey year” line, next to “pre-PCV10 (2012)”, “1 year post-PCV10 (2013)” … Also, I suggest adding the age of the subjects in each of the groups in the tables (“Infants (5-8 weeks)”, “Toddlers (12-23 months)”… and remove it from the footnote.

11. Table 1. Footnote. Please, explain in Material and Methods how were NT-lineages established.

Results

12. Line 165. “Survey year was negatively associated with overall carriage.”. However, percentage of carriage in the 3rd year was higher than in the 2nd year (p<0.001).

Discussion

13. Lines 234-242. In this paragraph the authors resume the main findings of their work. As they have been already described in the abstract and in the results section I recommend considering the possibility of deleting this paragraph. In the same way, I recommend to delete all the figures of “OR and P” through the discussion, to make easier its reading.

14. Line 262. “Unlike our study, however, both the Greenland and Alaska studies occurred in high income settings, which may comparatively reduce the impact of ethnicity based socio-environmental differences that might be related to pneumococcal carriage.”

If in the studies of Greenland and Alaska the carriage ORs were adjusted to other factors (as number of children <5 years living on the house, smoking, and other related socio-environmental factors) the statement would be speculative.

In the same way as the authors have studied the interaction of survey year and vaccination with ethnicity, maybe they should have also analysed if the two different ethnic groups of their study had different socio-environmental conditions (Young children living in the same house, low incomes, exposure to smoke, …) to see if ethnicity by itself was associate to carriage or were these conditions the responsible for the increased carriage.

15. Line 301. “…lack of selection pressure towards an overall increase in non-PCV10 carriage early post-PCV10 introduction, due to serotype replacement occurring only in iTaukei infants and toddlers, rather than across all participant groups [22].

However iTaukei infants had not received PCV10, so a direct effect of vaccination on serotype replacement (increase in non PCV-10 serotypes) was not expected. And if serotype replacement in iTaukei infants had been a consequence of the indirect effect of PCV10 on, iTaukei caregivers would had also suffered this serotype replacement…

16. Line 325. Limitations. From my point of view another limitation of the study are the data on the carriage density because of diferent factors. First, it can greatly depend on the sampling. Nasopharyngeal sampling is relatively uncomfortable and sampling in older children and adults can be more complicated than in younger children because the accession to the nasopharynx is more difficult. Besides, PCR can detect non-viable bacteria. Other studies have shown that there are serotype-dependent variations when detecting pneumococcal load (Messaoudi M, et al. PLoS One. 2016;11:e0151428). Other studies have shown a specificity of lytA PCR for detection of pneumococcal carriage of 75.9% (Gritzfeld JF, et al. Clin Microbiol Infect. 2014;20:O1145-51). I suggest that, although real time PCR is a good technique for measuring pneumococcal carriage especially in studies comprising thousands of samples, it has some limitations that have to be considered by the researches.

Reviewer #2: The manuscript by Neal et al addresses the factors associated with pneumococcal carriage and density in children and adults in Fiji, using data from cross-sectional studies that encompass the introduction of PCV10.

Pneumococcal carriage and carriage density were determined by qPCR targeting the lytA gene and serotyping was done by microarray. Factors associated with pneumococcal carriage and density were determined using logistic and quantile regressions, respectively. Variables included in the model were selected from an empirical univariate analysis and from an informed selection of variables. The latter were properly justified.

This paper is well structured and very well written. The methods are sound and the conclusions are properly supported by the results described. Some recognizable limitations to the study are acknowledged by the authors in the discussion section. All references are appropriate and were cited.

Please find below one minor comment:

Tables 2, 3 and 4: Each variable should include information regarding the number of samples that were excluded (presumably due to lack of information on that specific variable). The totals do not add up.

6. PLOS authors have the option to publish the peer review history of their article (what does this mean?). If published, this will include your full peer review and any attached files.

Reviewer #1: No

Reviewer #2: No

---

## [Author Response · Author response to Decision Letter 0]

9 Mar 2020

24 February 2020

Dear Dr Melo-Cristino,

We thank the Editor and the reviewers for their thorough assessment of our manuscript, and the detailed and insightful comments provided. A point-by-point response to editorial requirements, comments, and reviewer comments is below.

To enhance the reproducibility of your results, we recommend that if applicable you deposit your laboratory protocols in protocols.io, where a protocol can be assigned its own identifier (DOI) such that it can be cited independently in the future. For instructions see: http://journals.plos.org/plosone/s/submission-guidelines#loc-laboratory-protocols

Authors’ response 

We thank the Editor for this opportunity, however note that our laboratory methods are described in detail in our previous publications (Dunne et al Lancet Glob Health 2018; Satzke et al Vaccine 2019; von Mollendorf et al Vaccine 2019) and are in accordance with the overarching guidelines of the WHO (Satzke et al Vaccine 2013).

Authors’ response 

We thank the Editor and have amended our manuscript in line with the PLOS ONE style requirements

b) If there are no restrictions, please upload the minimal anonymised data set necessary to replicate your study findings as either Supporting Information files or to a stable, public repository and provide us with the relevant URLs, DOIs, or accession numbers. For a list of acceptable repositories, please see http://journals.plos.org/plosone/s/data-availability#loc-recommended-repositories.

Authors’ response

While our data has been fully de-identified, there remains an ethical restriction on making our data available publicly. The protocol approved by the Fijian Ministry of Health and Medical Services National Health Research and Ethics Review Committee, and by the University of Melbourne Health Sciences Human Ethics Sub-Committee, specified the purpose of the study and what the data would be used for. This information was conveyed during the informed consent process. Therefore, reasonable requests for data will need to include details on how data would be used, and will be subject to approval by the Fijian Ministry of Health and Medical Services National Health Research and Ethics Review Committee, which is the overseeing ethics committee for this study. This process considered the sensitivities regarding data from indigenous populations. 

"I have read the journal's policy and the authors of this manuscript have the following competing interests: CDN, CS, EKM, and EMD are investigators on a research projected funded by Pfizer for research in Mongolia. All other authors have no declarations of competing interests to declare."

Please respond by return email with your amended Competing Interests Statement and we will change the online submission form on your behalf.

Authors’ response

Thank you for the opportunity to amend this. We have responded via return email with the updated Competing Interests Statement, which now reads: “I have read the journal's policy and the authors of this manuscript have the following competing interests: CDN, CS, EKM, and EMD are investigators on a research projected funded by Pfizer for research in Mongolia. This does not alter our adherence to PLOS ONE policies on sharing data and materials. All other authors have no declarations of competing interests to declare.” 

Reviewers' comments:

Reviewer's Responses to Questions

Comments to the Author

1. Is the manuscript technically sound, and do the data support the conclusions?

Reviewer #1: Yes

Reviewer #2: Yes 

2. Has the statistical analysis been performed appropriately and rigorously? 

Reviewer #1: Yes

Reviewer #2: Yes 

3. Have the authors made all data underlying the findings in their manuscript fully available?

Reviewer #1: Yes

Reviewer #2: Yes 

4. Is the manuscript presented in an intelligible fashion and written in standard English?

Reviewer #1: Yes

Reviewer #2: Yes 

5. Review Comments to the Author

Reviewer #1: Manuscript Number: PONE-D-19-35241

Title: Factors associated with pneumococcal carriage and density in children and adults in Fiji, using four cross-sectional surveys.

Excellent work describing the factors associated with pneumococcal nasopharyngeal carriage and density in children and caregivers in Fiji, pre and post-introduction of PCV10. This article is a continuation of the previously published article of the same authors, and reference 22 of the present one: Dunne EM, Satzke C, Ratu FT, Neal EF, Boelsen LK, Matanitobua S, et al. Impact of 10-valent pneumococcal conjugate vaccine introduction on pneumococcal carriage in Fiji: results from four annual cross sectional carriage surveys. Lancet Glob Health. 2018;6(12):PE1375-E85.

The article is well written, methodology is, in general, sound (see comment 16 on pneumococcal density) and support the results obtained.

I only have some minor comments:

General comments

1. Vaccinated subjects in each age group are different: from non-vaccinated to different percentages of vaccinated depending on the year of the survey. However data have not been stratified by age-groups and analysed globally, so that the direct effect of vaccination on carriage might be somehow “diluted”.

Authors’ response

We thank Reviewer 1 for their time and consideration of our manuscript. We agree that the number and percentage of vaccinated subjects differs by age group, depending on the year of survey, and that the direct effect of vaccination on carriage might be diluted. However, the aim of the current study was to determine the factors associated with pneumococcal carriage in Fiji, taking into account the introduction of PCV10, so we explored the adjusted odds of carriage by potential risk factor. In our previously published paper, our aim was to measure the impact of PCV10 on pneumococcal carriage prevalence in Fiji, so we investigated adjusted prevalence rates by year post-PCV10 introduction and age group. We have not presented stratified analyses here, but these can be found in our previously published paper (Dunne et al Lancet Glob Health 2018). 

2. In many places of the article it is stated that “young age was positively associated with pneumococcal carriage”. However overall, PCV10 serotypes, and non-PCV10 serotypes carriage was lower in infants than in toddles and children. The authors should better specify to what age groups they are refereeing when talking of young age” as a general word.

Authors’ response

We have amended our manuscript for clarity

3. In the tables there is a P-value that in some items (survey year and participant group) is given in total for the other 3 categories. Please, explain if this P-value was calculated comparing the reference value (caregivers) with the addition of the results of all the other groups or if the comparison of each group with the reference one was the P-value expressed.

Authors’ response

In this study, we presented odds ratios and 95% confidence intervals for comparison between reference groups and other levels of categorical variables with multiple categories, such as survey year and participant group. The reported P-values for such variables were calculated to determine inclusion of the variable in the model, i.e., global P-values.

Detailed comments

Abstract

4. Line 23. I suggest adding “in October 2012” after “…post-introduction of ten-valent pneumococcal conjugate vaccine (PCV10).”

Authors’ response

We have amended our abstract accordingly.

5. May be the authors should differentiate in the abstract between factors positively or negatively associated with carriage, in order to be clearer for the reader. For instance, in line 26 it is stated that “Factors associated with pneumococcal carriage were years post-PCV10 introduction (global P<0.001) …” and in line 31: “…PCV10 carriage was also associated with PCV10 vaccination (0.58 [95% CI 0.41 – 0.82] P=0.002)”.

As PCV10 vaccination and survey year were negatively associated with PCV10 carriage (line 184) and survey year was negatively associated with overall carriage (line 165), but other factors were positively associated to carriage I recommend to write this clearly in the abstract to avoid possible misunderstandings.

Authors’ response

We have amended our abstract for clarity.

6. Abstract and end of discussion. Although the study is mostly descriptive, it would be desirable some conclusions of the study, not just the report of the data obtained.

Authors’ response

We have amended our abstract and conclusion to highlight the findings regarding ethnicity,

Material and methods

7. Children between 8 weeks and 12 months were not sampled, probably because they were in the vaccination period. However I would suggest an explanation of this decision in material and methods.

Authors’ response

The age groups were the same as reported for our study to measure the impact of PCV10 in Fiji on pneumococcal carriage prevalence (Dunne et al Lancet Glob Health, 2018). For the vaccine impact study, age groups were selected in order to best answer the primary research question based on those most likely to benefit from direct and indirect effects of PCV10; those likely to have the highest pneumococcal carriage prevalence; those age-eligible for PCV10 vaccination; and those most likely to transmit / be in contact with transmitters of pneumococci. The 5 – 8 week old infants were selected as they represent a vulnerable age group for pneumococcal disease, yet are too young to be vaccinated. The 12-23 month age group aligns with peak pneumococcal carriage and the majority of this age group would be fully vaccinated in the latter survey years. As such, the 5-8 week and 12-23 month old age groups were selected based on their potential to evaluate the indirect and direct effects of PCV10 introduction in Fiji, rather than a deliberate non-inclusion of children aged between 8 weeks to 12 months. 

This study used the same samples for convenience. For clarity, we have amended our manuscript to include a statement to this effect.

8. Line 141. Percentage of vaccinated subjects was different in each age-group (no vaccinated in infants and caregivers, 23% vaccinated children in 2015 and nearly 100% toddlers vaccinated in 2014-2015). In fact, 1105 (13.6%) participants had been vaccinated. Could these differences in the vaccination rates have biased the percentage of carriage in the different age-groups? The direct effect of vaccination in carriage could only be measured in 13.6% of participants, most of them in the 12-23 months age- group.

Authors’ response

In Dunne et al, Lancet Glob Health, 2018, our aim was to measure the impact of PCV10 vaccination and introduction on pneumococcal carriage prevalence in Fiji. In that study, we calculated adjusted prevalence rates by year post-PCV10 introduction, taking into account that there was no unbiased opportunity to measure direct effects alone, as vaccinated participants would be subject to both direct and indirect effects of PCV10. We found that prevalence of PCV10 carriage in both vaccinated and unvaccinated participants declined after the introduction of PCV10, with evidence of indirect effects in infants too young to be vaccinated and among older unvaccinated age groups. In this study, we aimed to investigate the host and demographic factors associated with pneumococcal carriage (such as age), not the direct effect of vaccination on carriage prevalence, so we explored the adjusted odds of carriage by potential risk factor.

9. Line 147. In the previous publication authors detailed that “19.8% of pneumococcal-positive samples contained more than one serotype.” I suppose that some of these samples contained PCV10 and non-PCV10 serotypes. In this case, to which group (PCV10 carrier or not) were assigned the subjects?

Authors’ response

In our revised manuscript, we have amended the Methods section to clarify that “detection by microarray serotyping of a PCV10 serotype and a non-PCV10 serotype from the same swab sample was recorded as positive for both PCV10 serotype and non-PCV10 serotype carriage (Dunne ED et al, Lancet Glob Health 2018). Any detection of a serotype by microarray was considered positive, regardless of relative abundance (Dunne ED et al, Lancet Glob Health 2018).

10. Table 1. I suggest to put in this table and in the following, the year in the “Survey year” line, next to “pre-PCV10 (2012)”, “1 year post-PCV10 (2013)” … Also, I suggest adding the age of the subjects in each of the groups in the tables (“Infants (5-8 weeks)”, “Toddlers (12-23 months)”… and remove it from the footnote.

Authors’ response

We have amended our manuscript accordingly. 

11. Table 1. Footnote. Please, explain in Material and Methods how were NT-lineages established.

Authors’ response

In our revised manuscript, we have explained that non-encapsulated lineages were classified based upon previously described genetic variants (Salter et al Microbiol 2012).

Results

12. Line 165. “Survey year was negatively associated with overall carriage.”. However, percentage of carriage in the 3rd year was higher than in the 2nd year (p<0.001).

Authors’ response

In our manuscript, we note that compared with the first year of survey (pre-PCV10) the adjusted odds of overall carriage were lower after the introduction of PCV10. We agree that the prevalence of pneumococcal carriage was notably low in the second year post-PCV10 introduction (2014) compared with other years. In addition, our 2015 sample collection period coincided with a local influenza outbreak (WHO FluNet http://apps.who.int/flumart/Default?ReportNo=7), which may have affected pneumococcal carriage prevalence that year. However, these details and discussion of them have been published by us previously in a paper regarding the impact of PCV10 on pneumococcal carriage prevalence in Fiji (Dunne ED et al Lancet Glob Health 2018); and the aim of this analysis was to determine factors associated with pneumococcal carriage, rather than analysis of prevalence rates. 

Discussion

13. Lines 234-242. In this paragraph the authors resume the main findings of their work. As they have been already described in the abstract and in the results section I recommend considering the possibility of deleting this paragraph. In the same way, I recommend to delete all the figures of “OR and P” through the discussion, to make easier its reading.

Authors’ response

We have amended our manuscript accordingly.

14. Line 262. “Unlike our study, however, both the Greenland and Alaska studies occurred in high income settings, which may comparatively reduce the impact of ethnicity based socio-environmental differences that might be related to pneumococcal carriage.”

If in the studies of Greenland and Alaska the carriage ORs were adjusted to other factors (as number of children <5 years living on the house, smoking, and other related socio-environmental factors) the statement would be speculative.

In the same way as the authors have studied the interaction of survey year and vaccination with ethnicity, maybe they should have also analysed if the two different ethnic groups of their study had different socio-environmental conditions (Young children living in the same house, low incomes, exposure to smoke, …) to see if ethnicity by itself was associate to carriage or were these conditions the responsible for the increased carriage.

Authors’ response

Previous analysis of pneumococcal carriage reported an ethnicity based disparity in pneumococcal carriage prevalence and invasive pneumococcal disease (Russell et al Ann Trop Paediatr 2006; Russell et al Pediatr Infect Dis J 2010; Dunne et al Lancet Glob Health 2018). Investigation of factors associated with pneumococcal carriage prior to the introduction of PCV10 found ethnicity to be an independent predictor of pneumococcal carriage (Russell et al, Paediatr Infect Dis J 2010; Neal et al Pneumonia 2014). Socio-demographic factors potentially associated with the ethnicity based disparity in pneumococcal carriage epidemiology in Fiji were explored in Neal et al Vaccine, 2019. In that study, we found indigenous iTaukei had larger household sizes (more people), more social contacts, and more frequent contacts of longer duration compared with Fijians of Indian Descent, but that observed differences in pneumococcal carriage prevalence by ethnicity were not explained by differences in socio-demographic patterns ethnicity. In the current study, our statistical models permitted evaluation of the association of ethnicity with pneumococcal carriage, adjusting for demographic factors such as number of young children living in the same house, low incomes, exposure to smoke etc., and found that ethnicity remained an independent predictor of pneumococcal carriage. 

We appreciate the reviewer’s suggestion, and as a result, explored possible interactions between ethnicity and sociodemographic factors. In our revised Discussion, we have indicated that we confirmed this, by checking the potential interaction between ethnicity and other variables included in models, with P<0.05 considered significant, and that found no evidence of interaction (as indicated by P values in Table 1 in our Response to Reviewers letter).

15. Line 301. “…lack of selection pressure towards an overall increase in non-PCV10 carriage early post-PCV10 introduction, due to serotype replacement occurring only in iTaukei infants and toddlers, rather than across all participant groups [22].

However iTaukei infants had not received PCV10, so a direct effect of vaccination on serotype replacement (increase in non PCV-10 serotypes) was not expected. And if serotype replacement in iTaukei infants had been a consequence of the indirect effect of PCV10 on, iTaukei caregivers would had also suffered this serotype replacement…

Authors’ response

We agree with Reviewer 1 that a direct effect of vaccination on serotype replacement among an unvaccinated cohort is not to be expected. Previously we have reported that in this early post-PCV10 setting, serotype replacement in carriage occurred solely in iTaukei infants and children, potentially due to the higher baseline carriage of non-PCV10 serotypes among infants and children, compared with adults (Dunne et al Lancet Glob Health 2018).. In our revised manuscript, we have clarified that serotype replacement in infants and toddlers rather than all age groups was the observed data, rather than speculation.

16. Line 325. Limitations. From my point of view another limitation of the study are the data on the carriage density because of diferent factors. First, it can greatly depend on the sampling. Nasopharyngeal sampling is relatively uncomfortable and sampling in older children and adults can be more complicated than in younger children because the accession to the nasopharynx is more difficult. Besides, PCR can detect non-viable bacteria. Other studies have shown that there are serotype-dependent variations when detecting pneumococcal load (Messaoudi M, et al. PLoS One. 2016;11:e0151428). Other studies have shown a specificity of lytA PCR for detection of pneumococcal carriage of 75.9% (Gritzfeld JF, et al. Clin Microbiol Infect. 2014;20:O1145-51). I suggest that, although real time PCR is a good technique for measuring pneumococcal carriage especially in studies comprising thousands of samples, it has some limitations that have to be considered by the researches.

Authors’ response

The methods used in detecting and quantifying pneumococcal carriage in the current study are globally accepted as the gold standard (Satzke et al Vaccine 2013). In addition, previous studies (including Gritzfeld CMI 2014) have shown a high degree of concordance with lytA qPCR and culture, and conducting viable counts on thousands of samples would be impractical.

Importantly, the lytA real-time PCR used in our study is highly specific for nasopharyngeal samples. However, we agree that specificity can be affected if oropharyngeal flora contaminate the specimen (e.g. during sampling) given the high abundance of closely-related streptococci in this niche. Indeed, we hypothesise that the lower specificity reported in Gritzfeld CMI 2014 in comparison with other studies, may be due to ‘contaminating’ oral flora in the nasal wash, consistent with the higher levels of alpha-haemolytic streptococci obtained from nasal washes (compared with nasopharyngeal swabs) in uninfected healthy adults (Gritzfeld et al., BMC Res Notes 2011). 

Detection of non-viable pneumococci may arguably be advantageous, but in any case was relatively uncommon in our study, where there were 2,456 swabs that were lytA positive, and of these, 2,395 (97.5%) were culture-positive. The paper by Messaoudi et al. is informative for the field, but its implications for our study are less obvious. The authors developed serotype-specific real-time multiplex PCRs and found that the nasopharyngeal density in hospitalised/pneumonia patients was around 3-fold higher for serotypes that were also isolates from the blood of the patients tested. They also reported that some serotype-specific qPCRs had different sensitivity, but the reason was unclear. The authors presented some reasons, including primer-probe interactions in the multiplex PCR, as well as differences in extraction efficacy between serotypes. Of note, our focus is on risk factors with overall, vaccine-type, and non-vaccine type carriage and density, rather than serotype specific carriage and density. We agree that there are likely some differences in the cell lysis between serotypes of pneumococci, but we are not aware of evidence that this characteristic differs between vaccine vs non-vaccine serotypes more generally. 

Reviewer #2: The manuscript by Neal et al addresses the factors associated with pneumococcal carriage and density in children and adults in Fiji, using data from cross-sectional studies that encompass the introduction of PCV10.

Pneumococcal carriage and carriage density were determined by qPCR targeting the lytA gene and serotyping was done by microarray. Factors associated with pneumococcal carriage and density were determined using logistic and quantile regressions, respectively. Variables included in the model were selected from an empirical univariate analysis and from an informed selection of variables. The latter were properly justified.

This paper is well structured and very well written. The methods are sound and the conclusions are properly supported by the results described. Some recognizable limitations to the study are acknowledged by the authors in the discussion section. All references are appropriate and were cited.

Please find below one minor comment:

Tables 2, 3 and 4: Each variable should include information regarding the number of samples that were excluded (presumably due to lack of information on that specific variable). The totals do not add up.

 Authors’ response

We thank Reviewer 2 for their reading and consideration of our manuscript. Our study included 8,109 participants. As noted in our Methods section, 38 participants who identified as “other” ethnicity were excluded from inferential analyses due to small numbers. A further 48 samples were excluded from microbiological analysis due to technical reasons such as insufficient volume, sample loss, and/or labelling errors. A further 61 pneumococcal positive samples were not able to be serotyped, because they were culture-negative (n = 51) or had low DNA yield from culture (n =6), or due to technical reasons (n = 4). In our revised manuscript, we have added footnotes to Tables 2, 3, and 4 for those variables where the pneumococcal carriage denominators do not total 8,023 (table 2) or 7,962 (Tables 3 and 4). 

6. PLOS authors have the option to publish the peer review history of their article (what does this mean?). If published, this will include your full peer review and any attached files.

Do you want your identity to be public for this peer review? For information about this choice, including consent withdrawal, please see our Privacy Policy.

Reviewer #1: No

Reviewer #2: No

We hope the revised version is now suitable for publication, and look forward to hearing from you in due course.

Yours sincerely,

Eleanor Neal, on behalf of all co-authors.

---

## [Editor Report · Decision Letter 1]

16 Mar 2020

Factors associated with pneumococcal carriage and density in children and adults in Fiji, using four cross-sectional surveys

PONE-D-19-35241R1

Dear Dr. Neal,

We are pleased to inform you that your manuscript has been judged scientifically suitable for publication and will be formally accepted for publication once it complies with all outstanding technical requirements.

With kind regards,

Jose Melo-Cristino, M.D., Ph.D.

Academic Editor

PLOS ONE
---

## [Editor Report · Acceptance letter]

18 Mar 2020

PONE-D-19-35241R1 

Factors associated with pneumococcal carriage and density in children and adults in Fiji, using four cross-sectional surveys 

Dear Dr. Neal:

I am pleased to inform you that your manuscript has been deemed suitable for publication in PLOS ONE. Congratulations! Your manuscript is now with our production department. 

With kind regards,

on behalf of

Prof. Jose Melo-Cristino 

Academic Editor

PLOS ONE